# Benchmarks as Microscopes: A Call for Model Metrology

**Michael Saxon**[♣]    **Ari Holtzman**[♠]    **Peter West**[♦]
**William Yang Wang**[♣]    **Naomi Saphra**[♥]
[♣]University of California, Santa Barbara  [♠]University of Chicago
[♦]University of Washington  [♥]Kempner Institute at Harvard University

## Abstract

Modern language models (LMs) pose a new challenge in capability assessment. Static benchmarks inevitably saturate without providing confidence in the deployment tolerances of LM-based systems, but developers nonetheless claim that their models have generalized traits such as reasoning or open-domain language understanding based on these flawed metrics. The science and practice of LMs requires a new approach to benchmarking which measures specific capabilities with dynamic assessments. To be confident in our metrics, we need a new discipline of *model metrology*—one which focuses on how to generate benchmarks that predict performance under deployment. Motivated by our evaluation criteria, we outline how building a community of model metrology practitioners—one focused on building tools and studying how to measure system capabilities—is the best way to meet these needs to and add clarity to the AI discussion.

## 1 Introduction

Just how good are our current language models? It's hard to say. Even the latest benchmarks are not scalable, relevant, or durable enough to predict performance in real-world settings.

Although engineering (Saka et al., 2024; Goyal et al., 2024), research (Bommasani et al., 2021; Bai et al., 2022), and policy (Cihon, 2019; NIST, 2023) decisions are grounded in the supposed capabilities of AI systems—particularly language models (Tolan et al., 2021; NIST, 2022)—even experts disagree about the nature (Li et al., 2023; Morris et al., 2023) and extent (Jumelet & Hupkes, 2018; Bender & Koller, 2020; Park et al., 2022) of these capabilities.

Most LM breakthroughs are judged either through *aggregate performance on narrow benchmarks* (Achiam et al., 2023) or *one-off manual analyses* (Bubeck et al., 2023). These assessments then inform conversations around scaling (Hoffmann et al., 2022), risk (Falco et al., 2021), and deployability. Popular static benchmarks inevitably **saturate** (Beyer et al., 2020; Ott et al., 2022) as each consecutive generation of models over-optimizes for performance on its evaluation sets—a process exacerbated by those datasets contaminating future training data—without resolving fundamental impasses over the nature of LMs or usefully informing their deployment. *We need benchmark practices that yield meaningful observations.*

> *The emergence of a new discipline: from homemade microscopes to optical metrology*
>
> In 1609, Galileo Galilei built one of the first optical telescopes. When he turned it around, Galileo found that he could also observe very small objects close up—and so microscopy was born (Singer, 1914). For centuries, these tools were built by the same scientists using them to make fundamental discoveries (La Berge, 1999).
>
> Eventually, the expertise required to design the precise tools to advance science outstripped scientists' glassworking skills. By the 20th century, large teams of specialists built orbital space telescopes (Leverington, 2012) and microscopes became mass-manufactured commodities (Davidson & Abramowitz, 2002). The science and engineering of measurement tools have coalesced into specialized disciplines.

---

Corresponding authors: `saxon@ucsb.edu, nsaphra@fas.harvard.edu`

At present, our LM evaluation practices resemble the state of astronomy and microbiology in the early 17th century—the same community analyzing the object of study (models) is also building the tools for that analysis (benchmarks). We believe that LMs require an independent professional and scientific community dedicated to building analytic tools just as microscope and telescope building have. Just as metrology—the science of measurement—coalesced from a useful skill for natural scientists into an independent discipline, we call for formalizing model evaluation research into a new field, **model metrology**.

First, we enumerate fundamental problems in language model assessment (**Section 2**). Popular benchmarks wrongheadedly attempt to measure *generalized capabilities*, a poorly defined goal (§2.1), distracting us from capturing real-world utility (§2.2). These issues and others, such as static benchmark saturation, are widely recognized, but have gone unsolved due to cultural disconnects between model builders, evaluators, and real-world users (§2.3).

From there we identify critical qualities that useful and concrete benchmarks *should* have (**Section 3**)—**constrained** settings, **dynamic** examples, and **plug-and-play** deployability—to motivate our proposals for rectifying the aforementioned issues, leading into our discussion of how a dedicated model metrology community is the way to provide them (**Section 4**).

Model metrologists can serve as a bridge between scientists, practitioners, and users (§4.1) and build benchmarks that meet our desiderata (§4.2). Within their field they will share knowledge, techniques, tooling, and theory (§4.3) to enable rigorous critique and auditing of other metrologists' work (§4.4), and advance the overall rigor of AI science. But how can we build the model metrology field?

**Section 5** introduces possible first steps, identifying existing disparate communities of domain-specific *proto-metrologists* and discussing how they may be organized into a unified community (§5.1). By soliciting real-world measurement needs (§5.2) and engaging with AI subfields that need better measurement practices (§5.3), the field can naturally grow.

**Section 6** concludes by noting the role metrology can play in high-level discussions around the fundamental nature of AI (§6.1), and how AI researchers—like the astronomers, phyicists, and biologists who came before them—might transition model metrology from an exploratory science, to a formal field, to finally, a mature engineering discipline (§6.2).

## 2   Problems with current benchmarks for LMs

While benchmarks have long driven AI progress, they are now used to support increasingly grandiose claims. When research communities believe that "solving" a benchmark represents core progress toward generalized intelligence, interest and investment naturally follow. Raji et al. (2021) document how the *common task framework*—public contests between systems assessed on common train and test sets (Donoho, 2017)—enabled advancements in concrete and tightly-scoped problems such as automatic speech recognition and machine translation, but has since been inappropriately extended to claim generalized capabilities in pretrained vision (Russakovsky et al., 2015) and language (Wang et al., 2019) models.

Current language modeling practices have shifted from training and testing on specific benchmark datasets to testing models in a zero-shot setting. Consequently, many researchers assume that *because* LMs aren't deliberately trained on task-specific train sets (Ge et al., 2023; Bai et al., 2024), performance on these benchmarks is *stronger evidence* of general capability than for fine-tuned models (Piantadosi & Hill, 2022; Mitchell & Krakauer, 2023). Though these assumptions are controversial—evidenced by the remarkably divergent perspectives of NLP and AI researchers (Michael et al., 2023)—the glamour of the promise of general intelligence has carried claims of generalized intelligence to a credulous public (Neri & Cozman, 2020), driving anxieties over AI risk (Ambartsoumean & Yampolskiy, 2023).

These attempts to assess general capabilities address a legitimate need: evaluation is important for guiding advances and comparing models (Phillips et al., 2000). Because modern LMs are used as *everything systems*, we may naturally wish to characterize their general capabilities (Morris et al., 2023). However, *when attempting to assess general capabilities, evaluations neither capture general competency nor predict performance on many downstream applications.*

> ### *On the wrasse fish & the pitfalls of generalized capabilities*
>
> As the apocryphal Einstein quote goes, "if you judge a fish by its ability to climb a tree, it will live its whole life believing that it is stupid" (O'Toole, 2017).
>
> Ironically, there is at least one example of fish intelligence outpacing primates, namely the economic puzzles solved in labs by the cleaner wrasse, a symbiotic species that lives in coral reefs and feeds on the parasites of larger fish. The fish are given meal options which should be eaten in a particular order due to variable reliability, and they find the optimal solution faster than capuchins, chimpanzees, orangutans, and even one researcher's four-year-old daughter (Salwiczek et al., 2012).
>
> The wrasse evolved to preferentially treat regular customers over reef visitors, tracking clientele across thousands of daily parasite cleanings (Gibson & Barnes, 2000). An artificial "wrassebot" would be ready to deploy only when it exhibits game theoretically-optimal strategies and "machiavellian" (Bshary, 2011) manipulations of clientele—but there's no need to solve grade school math problems or translate text.

## 2.1 Generalized capabilities are hard to define and contentious.

Narrow LM capability benchmarks are often derived either from tests for humans—e.g., GSM8K (Cobbe et al., 2021) and MMLU (Hendrycks et al., 2021)—or from *existing constrained benchmarks* for specific engineering problems in natural language processing such as question answering (Ho et al., 2023) or entailment recognition (Bowman et al., 2015). Performance on these benchmarks often predicts performance on similar test sets on the same task, to the degree that a violation of this expectation can be evidence of test-set contamination (Paster, 2023; Jain et al., 2024, fig. 5). However, claims regarding the real-world reliability of these systems are often unsupported (Liao et al., 2021). We believe these benchmarks cannot characterize broader capabilities, even when aggregated.

Critics of generalized capabilities benchmarks note that they lack **construct validity**—strong evidence that any evaluation represents a capability (O'Leary-Kelly & Vokurka, 1998) that they claim to measure (Davis, 2023). This discordance between metrics and resulting claims was already present for fine-tuned model evaluation (Raji et al., 2021) but has since worsened, as LM developers and their allies claim generalized intelligence (Bubeck et al., 2023), often based on a huge set of limited benchmark scores (Fei et al., 2022; Achiam et al., 2023). Similar claims are made for more specific abstract capabilities when researchers attribute benchmark performance to faculties like abstract reasoning (Yasunaga et al., 2021), language understanding (Moore, 2022), or common sense knowledge (Zhao et al., 2023b).

We are at an impasse. Though these claims of generality are contested (Murty et al., 2023), they are hard to conclusively disprove. Bender & Koller (2020) argue axiomatically that understanding cannot be acquired through the LM objective. Poor generalization across time (Lazaridou et al., 2021), tasks (Yang et al., 2022), and heuristics (Singhal et al., 2023) have also been provided as empirical counterarguments to LM capability claims. Given that humans struggle to even evaluate intelligence in other animals (De Waal, 2016), how can we assess slippery abstract notions like reasoning (Manning, 2022) in AI systems?

## 2.2 Benchmarks can aim for generality—or they can be valid and useful.

When benchmarks claim to test abstract capabilities, critics often question whether that capability is *necessary* or just *sufficient* for their solution (Potts, 2020). To substantiate a capability claim, a task must require said capability, but it may be impossible to prove that relationship. After all, both a studious human scholar and an answer key achieve 100% accuracy on an exam, but a piece of paper clearly does not possess the scholar's intelligence.

Consider a developer of a real-world application based on an LM, which we dub *builder-consumers*. For a builder-consumer, a useful benchmark must simply test if a system—regardless of abstract capabilities—is performant on their task. Meaningfully representing the deployment setting makes a benchmark **ecologically valid** (De Vries et al., 2020).

Are generality and ecological validity fundamentally in tension? Recent efforts to unify these goals such as HELM (Liang et al., 2023) provide a large collection of scores by harvesting existing benchmarks divided into specific scenarios (e.g., news domain tasks). By contextualizing these tasks within categories—and providing disaggregated scores over them—they aim to preserve task-level construct validity (Liao & Xiao, 2023) while capturing a "holistic" view of an LM's capabilities. Though many of these tasks may have construct validity, these holistic evaluation attempts do not capture generality—each benchmark represents a tiny view of a broader "task universe" (Liao et al., 2021). Ecological validity is particularly problematic when discussing "AGI-level" capabilities (Morris et al., 2023), though discussing the credibility of those notions is not central here.

Most model consumers are developing applications that rely on predictable LM behavior and therefore need to evaluate consistent, constrained capabilities for their domain. However, developers such as OpenAI (Achiam et al., 2023), Anthropic (Anthropic, 2023), and Google (Team et al., 2023; 2024) instead continue to focus on the same MMLU, GSM8K, and HumanEval test suites rather than on application-specific tests. These platforms provide API access to these models as a paid service, so *why aren't they benchmarking customer-relevant capabilities?* Maybe they are convinced that general intelligence is quantifiable by these benchmarks. Perhaps the deeper issue is that building bespoke evaluations is hard, and the domains are innumerable—will a collection of constrained tests ever be large enough to placate critics? Do scientists have a role to play in producing these gap-closing evaluations?

## 2.3 We know existing benchmarks are flawed. Why do we keep using them?

These criticisms are not novel—indeed, they're commonly expressed sentiments. Regardless, these flawed benchmarks remain dominant. Saturation is broadly acknowledged as a problem whose mechanisms present a fundamental challenge to benchmark validity. GSM8K, long used to assess mathematical reasoning, is fully saturated for "frontier" models—GPT-4 achieves near-100% accuracy with prompting and decoding tricks (Zhou et al., 2024a). We know models overfit even to hidden (but static) test sets (Gorman & Bedrick, 2019). We know that the long tail of incomprehensibly large pretraining datasets (Mitchell et al., 2022; Elazar et al., 2023) inevitably enables answer memorization (Alzahrani et al., 2024) or heuristic learning (Poliak et al., 2018; McCoy et al., 2019; Wang et al., 2021; Saxon et al., 2023) through similar examples (Peng et al., 2023; Kandpal et al., 2023). Poor construct validity is widely noted (Jacobs & Wallach, 2021), as is the futility of measuring generalized capabilities (Casares et al., 2022). So why do we still rely on these benchmarks? Perhaps:

1. Misalignment of interest/incentives for researchers and needs of users.
2. Fundamental difficulties in building benchmarks that meet our desiderata.
3. The allure of general intelligence attracts public, media, and investor attention.

As long as the LM community is the primary benchmark building community, these problems will persist. The incentives for academic researchers building benchmarks is 'impact,' as measured through citations and public use (Kang et al., 2023), and primary incentive for industrial actors is to demonstrate the superiority of their latest product. Unsurprisingly, 'top benchmarks' from a small number of elite institutions have become primary measures of AI progress (Koch et al., 2021). In the short term, a carefully-scoped and rigorously designed benchmark is as impactful as a well-hyped but soon-to-saturate one—but the former is much harder to make than the latter. The development of best practices for benchmark building cannot rely on the incentives of scientific machine learning research.

## 3 Qualities of useful, concrete benchmarks

The problems above can only be solved if we *abandon general metrics when making deployment decisions*. **Why study the wrasse fish's intelligence outside of the reef?**

We need a benchmarking culture and practice that empowers consumers to *specify their desired constraints* and *generate their own benchmarks*. Model producers should track progress by these scenario-driven evaluations. Good LM capability evaluations are:

**Constrained.** Benchmarks that characterize model behavior on a concrete task where domain experts can describe the boundaries a good system should stay within are *constrained*. Often, concrete problems can be defined in terms of verifiable rules. Given these boundaries as an objective, issues of scenario innumerability and ecological validity become much more tractable, as we can rely on domain expert understanding of real-world inputs.

**Dynamic.** A fixed dataset is easily memorized. To avoid stale metrics, we prefer dynamically generated test simulations where performance is measured by task-specific constraints.

**Plug-and-play.** Benchmark generating processes must be plug-and-play, i.e., run easily using customer-specific models or constraints. Expensive, one-off benchmarks are unlikely to foster open academic discourse—or provide value to downstream developers—and human preferences are expensive to gather and difficult to replicate. Accessible evaluation setups will entice domain experts to apply and even publish their constraints.

These desiderata are mutually reinforcing and enabled by *abandoning generality* and centering the needs of real-world builder-consumers and end users. Because model developers and AI researchers usually aim to improve general capabilities, focused real-world evaluation must instead rely on dedicated practitioners, i.e. metrologists.

## 4 The promise of a model metrology community

In this section, we explain how and why a dedicated discipline of model metrology can resolve the aforementioned issues, producing benchmarks that meet our desiderata and improve the state of LM production, use, and analysis. We lay out the benefits a dedicated metrology community would bring, the types of problems it would enable better solutions to, the research and engineering activities metrologists might undertake, and the cultural changes in broader AI science it would effect, discussing existing relevant work throughout.

### 4.1 A dedicated community can better connect researchers, developers, and users.

Given their role in spurring investment, flawed benchmarks and metrics lead researchers to waste time and effort developing methods that are ill-suited to any real-world application (Hoyle et al., 2021). Currently, LM benchmarking is disengaged from builder-consumers (Yang et al., 2023) and model-based service end-users (Xiao et al., 2024). Model metrology requires social change in both scientific and product communities to mitigate this disengagement. Even if tools and methods existed that could convert user constraints into quality dynamic benchmarks (see §4.2), enabling prospective LM-based application developers to successfully capture their use case in specifications may still be a challenge.

For example, consider the *Air Canada chatbot incident*. A Canadian court found Air Canada liable for paying a customer a nonexistent, off-policy bereavement discount promised by an LM-based customer service agent (Melnick, 2024). This agent, presumably using a GPT-based model, failed to adhere to policy—thereby failing as a customer service agent—despite GPT-3.5's near state-of-the-art performance on a massive suite of generalized benchmarks.

This incident might be described as the underlying LM lacking several different abstract capabilities. Perhaps the agent failed at *rule-following*, if the rules for bereavement discounts was specified somewhere in the system prompt context. Perhaps the agent failed in *common-sense reasoning*, if this specific policy wasn't explicitly provided but the list of all allowable discounts was. Regardless of the source of the error, it is unlikely that this specific failure case could have been predicted through generalized benchmark results.

However, a builder-consumer developing customer service agents could have tested for failures like this, but currently lacks the tooling to do so. Within the customer service agent domain, *common sense* entails not making promises that violate policy. A developer could manually enumerate every line of the policy, probing the agent for examples where it would fail. Model developers aren't thinking what abstract capability failures mean in diverse domains, *but domain experts who will use the models know what their needs are*. Model metrologist-developed methods will enable builder-consumers to plug their own constraints in to generate ecologically valid benchmarks for their specific task.

### 4.2 Metrologists will produce targeted dynamic benchmarks for complex problems.

How might a dedicated community build evaluations that meet our desiderata? Let's consider the Air Canada incident. Suppose a developer had a concrete list of rules for a customer service agent to follow, including not promising off-policy transactions. How do we use such rules to dynamically evaluate an LM for this constrained domain?

One technique could be to leverage an adversarial LM as a source of variation, generating many test cases attacking the task-domain constraints. In our airline customer service example, an LM could be prompted to generate role-play scenarios of various challenging customers: a child pranking the system, a client who struggles with technology, a jailbreaker looking for a big discount, or a panicked and angry stranded traveler.

Model outputs conditioned on these adversarial test dialogs could be judged deterministically against policy constraints, detecting issues like diversion from company policy. Even though the evaluation is dynamic, no human evaluator needs to manually enumerate all edge cases, as an expert has already fixed the deterministic rules. This style of evaluation exists—within the silo of LM security research (Shayegani et al., 2024; Zhu et al., 2023)—a metrology community would further its development, dissemination, and deployment.

Evaluation scenarios like this one are possible with current technology, as LM adversaries have been already been used for stress testing and assessment (Chan et al., 2024). There is mounting evidence that, using reversal, LMs can generate exemplars that they can't correctly respond to (Berglund et al., 2023; West et al., 2023). Given well-scoped constraints, model outputs can be evaluated deterministically, e.g., using variation between minimal pairs (Ribeiro et al., 2020) or fulfillment of a set of requirements (Hu et al., 2023), rather than using arbitrary and opaque LM-judgements of dubious reliability (Oh et al., 2024).

Though we have proposed a dynamic evaluation pipeline for one constrained setting, we are not claiming to have described the best way to produce a strong benchmark-generating process for all settings. Concerted research is necessary to develop best practices for metrology. Professional model metrologists will have to be competent at developing, formalizing, and sharing insights from disparate benchmark efforts for the benefit of the field.

### 4.3 Model metrologists will establish shared knowledge & techniques.

Even as we develop increasingly sophisticated evaluation methods, our community lacks consensus on their validity and best practices for their use. Metrics that use automatic scores from reference-similarity (Kocmi & Federmann, 2023) or correctness (Wang et al., 2023a; Mizumoto & Eguchi, 2023) are hotly debated (Chiang & Lee, 2023). For metrologists to use a technique with confidence, they need community consensus on its efficacy. We need:

**Shared framings of abstract capabilities across concrete settings.** Although abstract capabilities like "reasoning," "understanding," or "rule-following" are ill-defined and unquantifiable in general, they can be used to frame desired behavior and edge cases to avoid in constrained settings. For example, the apparent lack of rule-following exhibited in the Air Canada incident suggests that constraint-based adversarial agent testing may uncover failure modes in a customer service chat bot setting. Techniques developed in pursuit of that evaluation could probably be leveraged for many other rule-following problems in other constrained settings. Transfer of this knowledge would be facilitated by having dedicated metrologists—rather than customer service chat bot developers—building and promoting these constrained evaluations. By comparing the results of a method deployed across disparate settings, metrologists will guide further tool development.

Evidence Centered Benchmark Design (ECBD) (Liu et al., 2024) is an example of proto-metrology work toward this direction. They lay out a framework for assessing whether a benchmark actually captures a desired (often abstract) capability through analysis of specific "test items" within a target application context. Effort in using a standardized framework to describe a capability in one domain may transfer to another, and through the accumulation of evidence on how these framings perform in the wild, metrologists will develop a more sophisticated vocabulary to develop the practice of model measurement and assessment.

**A shift from observations to theories and science.** The AI community is sharing observations about LMs too fast for any one researcher to follow. Without replication and meta-analyses, scientists cannot determine which observations expose meaningful patterns and which represent random idiosyncrasies. For example, the evidence connecting benchmarks on various domains is weak (Fergusson et al., 2023). Although different math tests are usefully correlated (Paster, 2023) and small test sets of under 100 samples estimate on large static multi-task benchmarks (Polo et al., 2024), the future of these approaches remains uncertain. Can we predict the limits of this generalization? Will it hold for new classes of models? Will it hold for dynamic benchmarking or constraint-based benchmarks? A dedicated discipline could synthesize the growing evaluation literature by replicating, analyzing, and eventually canonizing findings into useful *theories of metrology*.

> *The luminiferous aether & advances in science driven by surprising measurements*
>
> When 17th century physicists first proposed that light is a wave, they posited that it must travel through a physical medium, which they dubbed the *luminiferous aether*. As the earth moves through space, the theory predicted that an *aether wind* would be observed, making the speed of light on earth different in different directions.
>
> However, the aether remained unobservable until the late 19th century invention of the *interferometer*, an instrument to measure light interference patterns. The Michelson-Morley experiment, intended to demonstrate the direction of the aether wind by comparing the speed of light in orthogonal directions, failed to find any differences. **Enabled by advances in measurement technology**, this "most famous failed experiment" (Blum & Lototsky, 2006) revolutionized 20th century physics, ultimately giving rise to relativity and quantum theory (Shankland, 1964).

By producing new measurement tools, model metrologists can not only validate existing models but drive LM science as a whole. As in other sciences, better measurements can precipitate questions that our current scientific paradigms are not yet capable of asking (Kuhn, 1962). For example, improved metrics can enable more sophisticated testing of scaling laws (Schaeffer et al., 2024) and discover associations between specific capabilities and error types. While it is impossible to predict what future paradigm shifts will look like, we are confident that surprising yet high-confidence observations—which metrology is intended to enable—will have an important role to play.

**Quality benchmark-building tools.** In the long term, metrologists should aim to develop tooling for automated benchmark generation by domain experts who are not necessarily evaluation experts, as discussed above. This will require technical innovations such as methods to expand a high-level task description into a set of exemplars, or prompting an LM to behave adversarially against a task-specific LM system. Among other tools, metrologists must develop prompting techniques that test the boundaries of rule-following in one setting (e.g., customer service) and generalize better to other settings (e.g., planning navigation). These concretely motivated—but generalized—techniques would be more appropriate than those explicitly designed for reasoning assessment. Perhaps the best way to test rule-following is to monitor agents interacting with simulated situations. Proper evaluation, however, requires both creativity and broad knowledge. Metrologists could stress-test chat bots by eliciting interactive personae by prompting (Cheng et al., 2023).

Task Me Anything (Zhang et al., 2024) is a recent example of work toward automated evaluation that is accessible out-of-the-box to non-experts. The authors propose a technique to build a multimodal LM evaluation set to answer specific queries about language model capabilities such as "which model is best at counting objects?" by selecting samples from existing static benchmarks. Techniques such as this coupled with sample generation strategies could be shared between metrologists across application domains in combination with other tools such as automated sample generation to build better dynamic benchmarks.

While some tasks are well-suited for constraint-based evaluation, for others (such as general purpose chat agents) the target is human preference. The gold standard for human preference evaluation in interactive LM applications are *competitive interactive evaluations*

such as Chatbot Arena (Chiang et al., 2024) where multiple systems are compared head-to-head on genuine human-generated queries. Although "chat agents" are not a particularly constrained domain, these evaluations do capture genuine human preference dynamically. Unfortunately, they are expensive to run and inconsistent, as any new system will be run head-to-head against all other models on new human interactions that must be collected over time. In light of these shortcomings *LM-as-a-judge* techniques have been proposed, where an evaluator language model is used as a proxy for human preference feedback (Zheng et al., 2023). Metrologists might expand this technique beyond chat bot evaluation.

### 4.4 Metrology culture prioritizes data work, methodological rigor, and proactive criticism.

Despite its importance, data work is devalued in AI research communities compared to more glamorous theoretical, empirical, and modeling work (Sambasivan et al., 2021). Even without a cultural shift in AI research, a metrology-focused community should center data and benchmarking work as first-class contributions.

Recent existing work has promoted rigor in benchmarking by identifying model cheating on benchmarks (Chen et al., 2024), finding errors in dynamic benchmark generators (Saxon et al., 2024a), and producing meta-metrics to find benchmark failure modes (Saxon et al., 2024b). However, these efforts are *reactively* responding to flawed work, rather than *proactively* identifying best practices for capability measurement. A metrology community should aspire to be the latter.

## 5 How do we build the model metrology discipline?

Having established the motivation, purpose, and necessity of the dedicated discipline of model metrology, we now discuss ways we might build the field. Because an evaluation discipline is not part of the existing language modeling community's culture—including among model consumers—metrology can only be built alongside substantial social change.

### 5.1 Uniting proto-metrology communities

Many ML, NLP, and AI conferences already hold benchmarks and evaluation tracks. Work on metric development and benchmarking best practices has regularly appears at ICLR (Lu et al., 2024), *ACL (Maynez et al., 2023), CVPR (Xu et al., 2022), and NeurIPS (Zhang et al., 2023). These researchers are effectively *proto-metrologists* establishing foundations for this field. As a starting point, current and aspiring metrology researchers should be familiar with these pioneering works. Workshops and evaluation-focused venues could facilitate cross-engagement between aspiring metrologists and ultimately provide an intellectual home. For inspiration, we look to existing communities of proto-metrologists concentrated in domain-specific venues (Saphra et al., 2024).

The machine translation (MT) community has invested considerably in rigorous benchmarks and metrics. The Conference on Machine Translation (WMT) (Kocmi et al., 2023) has run shared tasks that *simultaneously benchmark translation systems and translation quality metrics* since 2006 (Bojar et al., 2016). With buy-in from both academic and industrial researchers developing MT systems, the annual WMT shared task evaluates MT systems and metrics on new language pairs and domains every year. In so doing, the MT community continually refreshes their measurement practices as the field advances. Those MT researchers focused primarily on building quality evaluation metrics are effectively *MT metrologists* already.

Similarly, the Generation, Evaluation & Metrics (GEM) Workshops (Gehrmann et al., 2023), focused on advancing and evaluating text generation systems for specific tasks like data-to-text and summarization, are another good example of a proto-metrology community. Alongside shared tasks on building these text generation systems, they emphasize research on metrics for evaluating generated text against gold references to better compare competing submissions. Their GEM benchmark was an early attempt at building a protocol for living text generation assessment where new tasks and metrics could be slotted into a comprehensive benchmark over time (Gehrmann et al., 2021).

Strong proto-metrology work is continuously being published—both within these communities and at larger venues—yet knowledge transfer between these proto-metrologists is scattershot. Topic modeling researchers have assessed how different topic model quality metrics vary considerably across corpus domains (Hoyle et al., 2021), yet despite its general transferability, this finding is confined to the topic modeling community. Text-to-image researchers have invented intricate ways to generate directed graphs of requirements to check prompt-image faithfulness (Cho et al., 2023), yet outside this community, these techniques have not disseminated. The move from annual static competitions (Harman, 1993) to dynamic, modular packages (Thakur et al., 2021) has greatly advanced the practical state-of-the-art in information retrieval—but researchers outside IR may be unaware.

Right now we effectively rely on happy accidents to spread evaluation knowledge across these disciplinary barriers. By organizing this work not as "[domain]'s evaluation research" but as *core model metrology research applied to* [domain], we can render this knowledge transfer routine. Model metrology venues should bring the culture of WMT and GEM to evaluation writ large, so lessons from all these domains can be shared—to start we should collect and promote these disparate threads of capability measurement research as a cohesive whole.

## 5.2 Soliciting novel constraints and edge cases to benchmark

A model metrology community must be built by engaging with domain experts and model consumers. Metrology practitioners can use case studies such as the Air Canada incident as starting points to experiment with constrained benchmark-generating systems, but ultimately *useful constraints can only be provided by domain experts* who know the boundaries of their tasks. Academic metrologists and metrology venues should actively solicit specifications for new tasks. These expert-designed settings could be framed as shared tasks or even as concrete evaluation bounties. While we expect that industry and nonprofit customers developing LM-based applications will happily contribute their constraint specifications, as they stand to benefit most when their needs are prioritized by model training institutions, incentivizing academic researchers to invest in this work may prove challenging.

After all, data-focused conferences such as LREC already publish specialized training and test sets for constrained-domain tasks—but our academic incentive structures based on 'prestige' and technical 'novelty' devalue these venues compared to general AI venues like NeurIPS. A metrology community may effect sociological change toward valuing this grounded and concrete work by providing opportunities for genuine technical novelty (eg., benchmark generating processes) alongside crucial data work on builder-consumer-relevant constraints. Practical model metrology will probably emerge as a profitable industry where consultants operate similarly to penetration testing teams in information security, customizing stress tests for each scenario based on their broad knowledge of evaluation.

## 5.3 Engaging with related fields

At its start, model metrology draws on the evaluation work already taking place in many domains of machine learning, artificial intelligence, and natural language processing research. Though organizing this work in a cohesive community has many benefits, model metrology will only succeed if it continues to closely engage with its parent communities—but those communities stand to benefit from a stronger metrology culture as well.

Empirical research requires measurable dependent variables. Beyond building better benchmarks, model metrologists will be well-positioned to build *ecologically valid observational tools* to inject rigor into capabilities-focused empirical research. In particular, we anticipate black-box model analysis (Belinkov et al., 2023), human-computer interaction (Liu et al., 2023), robustness (Zhong & Wang, 2023), and interpretability (Räuker et al., 2023) research will benefit from engagement with metrologists.

While it is widely accepted that model metrics improve at larger scales, the metrics chosen are often critiqued. Scaling laws have been claimed on LM loss (Hoffmann et al., 2022), static benchmark performance (Srivastava et al., 2023), or—in the extreme case—a completely decontextualized *y*-axis simply labeled "intelligence" (Anthropic, 2023), yet these metrics

cannot fully support popular claims about scaling laws in general intelligence or practical utility. A different test set and metric can elicit inverse (McKenzie et al., 2023) and U-shaped scaling curves (Wei et al., 2023) with respect to model size, demonstrating a need for serious discussion of measurement methodology led by metrologists.

# 6 Conclusions

The time has come for disparate threads of research in LM benchmarking and evaluation scattered across many domains to coalesce into a cohesive model metrology community.

Investments into techniques to build constrained, dynamic, and plug-and-play evaluations to end user or builder-consumer specifications will enable more granular and concrete model evaluation. Adoption of these granular evaluations at scale will enable model makers more holistically argue for the utility of their LMs over competitors.

A community focused on these best practices for measurement and assessment will in turn be able to leverage this knowledge to identify new experimental directions into deeper model capabilities. If successful, the agenda will operationalize many areas of interest within AI, including alignment, fairness, common sense, knowledge, and reasoning. These nebulous objectives can be freed from the innumerability and construct validity issues they acquire when treated as generalized open-domain capabilities.

Metrology will produce real-world applicable evaluation techniques to help LM users make informed decisions and help model developers track incremental progress. We believe grounded progress benchmarks will also improve public discourse around AI.

When benchmarks attempt to simultaneously track core scientific progress and product effectiveness, they fail to achieve either. A dedicated evaluation discipline will be able to make great evaluations for each goal with shared methodology.

## 6.1 Model metrology & the artificial general intelligence (AGI) discussion

Present benchmarking culture suffers from conflicting efforts to practically compare models and to produce evidence in debates about AGI. Despite many high-profile cases of proven benchmark contamination, AGI optimists continue to equate improvement on static benchmarks with increases in robust intelligence. In their framing, critics are "moving the goalposts" by dismissing newly saturated benchmarks. Meanwhile, skeptics spin underperformance on a specific benchmark as evidence that a target capability is impossible for LMs. As each benchmark saturates, the cycle of hype and deflation continues and little is learned.

We believe our vision for metrology is useful as it directly gets at a core reason most people actually care about the AGI: the promise of making drop-in replacements for humans in specific jobs. If this really is what we care about, why not measure it directly? Constrained and ecologically valid benchmarking can finally decouple practical evaluation from ideological arguments about AGI.

A culture of model metrology will hopefully drive everyone to make *weaker statements* about intrinsic model capabilities grounded in quantifiable real-world capacities. These evaluation experts can promote a healthier, calmer public-facing discourse around AI.

## 6.2 The end game: model assessment *without* a model metrologist

In the best-case scenario, the model metrology agenda succeeds by building a mature engineering discipline and making standardized off-the-shelf benchmark-generating processes a commodity. Our proposal is modeled on the history of microscope manufacture. For most applications requiring a microscope, the exact desired instrument is already mass-produced. For truly niche applications (e.g., assessing semiconductor deformation) custom-built metrology solutions are still needed (Houghton et al., 2016). One day, LM consumers should likewise meet their measurement needs out-of-the-box without hiring a metrologist. This independence is the ultimate goal of a model metrology discipline.

## Acknowledgements

Thank you to the countless colleagues who have discussed the problems of evaluation with us and who have made public statements on them, inspiring and informing this work.

Thank you Greg Durrett, Tejas Srinivasan, Leif Hancox-Li, Prajjwal Bhargava, John Kevin Cava, and our anonymous reviewers for reading and providing feedback on earlier drafts.

This work was supported in part by the National Science Foundation Graduate Research Fellowship under Grant No. 1650114, and CAREER Award under Grant No. 2048122. This work was enabled in part by a gift from the Chan Zuckerberg Initiative Foundation to establish the Kempner Institute for the Study of Natural and Artificial Intelligence.

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

## Limitations *(and rebuttals)*

**You overlooked existing dynamic/constrained/construct valid benchmark examples!** Current automatically-generated benchmarks are largely confined to toy problems of limited interest to practitioners. For games like chess (Feng et al., 2023) and reasoning and planning problems from block worlds (Valmeekam et al., 2022; Stechly et al., 2024), test examples can be generated and evaluated deterministically. Programming is a notable exception— it is commercially relevant, deterministically evaluable, and test examples are relatively easy to dynamically generate (Allamanis et al., 2024). However, even for programming, benchmarks over fixed problem sets (e.g., HumanEval (Chen et al., 2021)) reign supreme. Clearly further emphasis on the necesity of dynamic benchmarking is needed here. Dynamic evaluations of LM coding capabilities such as LiveCodeBench (Jain et al., 2024) do source real coding problems from the internet, but this approach is difficult to transfer to other domains. Metrologists will create more dynamic benchmarks reflecting real applications.

As for constrained and ecologically valid benchmark examples we didn't discuss, WildBench (Lin et al., 2024)—a static benchmark built atop exemplars collected from WildChat (Zhao et al., 2023a)—is one rare example of a benchmark that truly is representative of its target distribution of real user dialogue. However, it still is static, and chat agents writ large are not a very constrained domain. This is the reason that the LMSys benchmarks fail to meet all our desiderata (Chiang et al., 2024). Furthermore, the actual users testing these systems are largely LM enthusiasts or researchers themselves—this strains the ecological validity of most arena-style benchmarks.

**LMs evaluating LMs? How can a system measure capabilities we don't know it has?** There are risks to relying on the target of analysis to self-verify. After all, how can we use a model to measure its own capabilities (or those of similar models)? One solution is to expand out a set of objectively verifiable characteristics with an LM, checked externally.

Preliminary evidence suggests that even GPT-4 fails to match human annotator performance in open-domain claim verification (Wang et al., 2023b). If we prompt a model to generate sentences, then ask GPT-4 to evaluate the generated text, how can we trust that GPT-4's judgements capture anything meaningful? These LM judgements are also problematic for

evaluation because they are not even comparable, as different models produce substantially different outcomes (Zhou et al., 2024b).

> ### *Reflections on trusting trust*
>
> Ken Thompson's Turing award acceptance speech, "Reflections on Trusting Trust" (Thompson, 1984), details how he hid a backdoor Trojan horse in early source versions of a C compiler. Because the compiler was bootstrapped, i.e., new versions of a compiler were compiled by the previous version, this backdoor was nearly impossible to detect or remove without being aware of its introduction.
>
> The backdoor was included even in versions of the compiler binary built from source code without the backdoor, as long as that source was compiled using a binary descended from Thompson's modified code. His discovery, shocking in 1984, seems almost mundane today: *If any part of a complex system is compromised, the entire system is compromised.* For modern automated metrics employing blackbox language models for their own evaluation, verification, and even training, we must view each element as potentially compromised by the flaws in proprietary models.

**Even dynamic benchmarks will go stale**  Living benchmarks based on arbitrary rules can be gamed by exploiting the idiosyncrasies of the supervising model and discrepancies in the simulation environment. We have no visibility into the decision processes and therefore no real guarantees of its validity (Oh et al., 2024). Therefore, developing a living metrology community is crucial. Researchers and practitioners will need to refresh their benchmark generators with new methods. Where generative techniques become obviated, breakthroughs in measurement can occur.

