# OpenReview forum: "Benchmarks as Microscopes: A Call for Model Metrology"
_colmweb.org/COLM/2024/Conference — COLM_

### Official Review · Reviewer_nguQ · 2024-05-10

**Rating:** 7
**Confidence:** 3
**Ethics Flag:** 1

**Summary:**

This paper has a relatively high quality, as the authors construct a compelling case against current large language model (LLM) benchmarking practices, meticulously exposing their flaws and providing clear explanations for their shortcomings. The inclusion of real-world examples, such as the Air Canada chatbot incident, effectively illustrates the consequences of relying on flawed evaluations. By introducing "model metrology" as a novel discipline dedicated to LLM evaluation, the authors offer a persuasive solution to a pressing problem.

The paper's clarity is good enough. The logical and coherent structure of the argumentation enables readers to effortlessly follow the authors' reasoning, while precise definitions of key terms ensure a unified understanding. As a result, readers can gain a deep understanding of the challenges inherent in LLM evaluation and fully appreciate the proposed solutions.

The concept of "model metrology" is genuinely novel and sets this work apart from others although i am not an expert in this field. By challenging the status quo, the authors advocate for more ecologically valid and dynamic evaluation methods, such as using adversarial LMs to generate diverse test cases. This innovative approach has the potential to revolutionize LLM assessment and positions the paper as a contribution.

This work addresses a critical issue in AI: the lack of reliable and informative LLM evaluation methods. By proposing the model metrology framework, the authors provide a path towards a more nuanced understanding of LLMs, ultimately leading to responsible development and deployment. The emphasis on collaboration between researchers, developers, and users is crucial for building effective benchmarks and advancing the field. Overall, this position paper is promising for LLM evaluation, offering a fresh perspective and innovative solutions to a pressing challenge.

**Reasons To Accept:**

1. The paper is well written overall and makes it easy to understanding.

2. The introduction of "model metrology" offers a novel and innovative approach to LLM evaluation.

**Reasons To Reject:**

1.  The paper could benefit from providing more concrete examples or case studies of how these benchmarks could be designed and used in practice.

2. The paper proposes the development of dynamic benchmarks, but it doesn't delve deeply into the practical challenges of creating and implementing such benchmarks.

---

> ### Author Rebuttal · Authors · 2024-05-29
>
> Thank you for your positive review. We are happy that you appreciate our argument! We agree that modern benchmarking has critical issues that must be addressed, and hope that our work starts a conversation that can lead to benchmarks that serve both as scientifically rigorous observational tools and meaningfully enable engineering decisions.
>
> Regarding your critiques:
> - We agree that our paper could benefit from more concrete case studies. With the extra CR page and the space we will be reclaiming by editing for brevity, we plan to add several case studies of flawed approaches (LMSys, specific stale datasets, HELM) and one example of how this approach might work. In order to give a positive example, we will expand the Air Canada case study by detailing how problems could have been detected during evaluation. For example, an evaluator could use prompts to generate a number of common and extreme but realistic cases to initiate dialogs, which would be evaluated by a second system to evaluate those dialogs based on constraints on what claims an agent can legitimately make.
> - While an extended discussion is outside of our scope, we plan to discuss some of the issues deployment-specific evaluations pose, as you suggest. One issue is that of creating general guidelines, as it is challenging to transfer lessons from the design of a deployment-specific evaluation to any other scenarios. Another is the challenge of evaluating evaluation: If these metrics are designed for a potentially proprietary deployment scenario, how can we iterate on real world outcomes to settle on best practices for the field? Given the extra space provided by the CR and extensive editing for brevity, we expect to be able to discuss these issues in detail.

---

> > ### Comment · Reviewer_nguQ · 2024-06-04
> >
> > I've read the author response, and don't have much to change for my review.

---

> > > ### Author Response · Authors · 2024-06-04
> > > **Thank you for your response**
> > >
> > > Thanks for the response and acknowledgement! Your review's helpful advice will strengthen our final draft.

---

### Official Review · Reviewer_3utb · 2024-05-11

**Rating:** 8
**Confidence:** 4
**Ethics Flag:** 1

**Summary:**

This paper offer a critique the current paradigm of benchmarking LMs. They identify several issues with the current paradigm, including benchmark saturation, lack of ecological validity, an over-focus on benchmarking generalized capabilities. The authors make a case for a new field of study - model metrology - that focuses on studying and designing LM measurements.

Overall, this paper is well written, well researcher, and well motivated. Parts of the papers argument have been made elsewhere, but the authors bring together a broad set of literature in a well organized may to make a compelling case for concentrated focus on LM measurement.

**Questions To Authors:**

Benchmarking is one particular paradigm of evaluation, not not the only one. How do you see other evaluation efforts fitting into the proposed new field of model metrology?

Minor typos:
- Sec. 4.2: "In this section we these questions.."
- Sec 7: "If we use prompt a model to generate sentences..."

Relevant literature:
- Jacobs & Wallach 2021. "Measurement and Fairness"
- Koch et al. 2021. "Reduced, Reused and Recycled: The Life of a Dataset in Machine Learning Research"

**Reasons To Accept:**

This is a well researched and well cited paper that presents a compelling and timely argument relevant for COLM.

The general arguments are all compelling: that the field should move towards constrained and ecologically valid measurements, invest in methods to enable dynamic benchmarking, and formalize a specialized and rigorous field of study to advance LM measurement.

The paper appropriately recognizes expertise outside the field of ML as being essential to building robust and ecologically valid benchmarks and evaluations.

**Reasons To Reject:**

Benchmarks are but one way of evaluating LMs.. the paper could be strengthened by engaging with other forms of evaluation and/or discussing elements of a user's experience/model capabilities that cannot be fully captured through the benchmarking paradigm.

---

> ### Author Rebuttal · Authors · 2024-05-29
>
> Thank you for your thoughtful response! We agree that this discussion is very timely, and we look forward to integrating suggestions from all our reviewers into the final paper.
>
> Our suggested approach would, as you suggest, be combined with other methods of evaluation. Aggregate benchmarks and LMSys-style tests of versatility are important for judging whether a general purpose model is good enough to even consider for downstream applications. Evaluations based on adversarial settings or bias detection are also necessary well before deployment. After evaluating a model’s potential for deployment, we also may want to combine metrology with traditional quality assurance testing and user studies.
>
> Thank you for noting the typos and additional references!

---

### Official Review · Reviewer_GTUZ · 2024-05-11

**Rating:** 6
**Confidence:** 3
**Ethics Flag:** 1

**Summary:**

This position paper proposes a new discipline: "model metrology", aimed at generating new benchmarks for Language Models (LM) evaluation that overcome the limitations of current benchmarks (saturation, memorisation, non-constrained scope, etc.). First, the paper analyses current limitations of current benchmarks, then proposes a number of desired features for such a new generation of benchmarks: constrained, dynamic, and plug-and-play. Finally, it analyses the characteristics of the proposed "model metrology" discipline and proposes a roadmap to achieve such a vision.

The topic is a relevant and interesting one. Limitations of current benchmarks for LLM evaluation are well known, and there are active research efforts in the direction of mitigating them, to which this work could contribute. The paper contains interesting ideas (the idea of model metrology as a new discipline is appealing) and can be the basis for stimulating a debate in that direction.

However, the paper has some flaws in my view. In general, there are a number of good ideas, but some of them not enough elaborated, some of them are bit vague (e.g., section 4.3) and some others not so novel, such as the "constrained" feature (task-based and extrinsic evaluations have been around for decades). In contrast, the "dynamic" aspect is a more contemporary need in NLP. However, I miss the "human in the loop" aspect as well.

In the paper, there are too subjective claims such as "[OpenAI, Antrophic, Google] still cling to the idea that general intelligence is quantifiable by these benchmarks", which are not sufficiently supported. (Do these vendors really think so? If so, citations would be welcome).

As an example of the limitations of current benchmarks, the Air Canada Chatbot incident is presented. In fact, as stated by the authors, is "implausible that this failure case could have been predicted through generalized benchmark results". However, this does not contradict the value of such benchmarks to allow testing (and improving) the model along several specific dimensions, before integrating them in a final system. The tests of the system as a whole (model + policies specification + UI, etc.) in the Air Canada case were obviously incomplete, but this is possibly because good test engineering practices were not followed, not invalidating the current LLM benchmarks per se.

From a stylistic point of view, the writing style is a bit wordy at times and difficult to follow, lacking the conciseness of academic writing. Some grammar constructions are a bit weird, for instance in section 3: "One reason [for what?] is that static benchmarks of that form [which form?] are easily memorized, […] “, or in "outlook and roadblocks to this discussed in §4.2", where the auxiliary verb is missing. There are also many contractions along the whole text (aren't, it's, can't), which is unusual in academic written English.

Despite the extensive references section, there are some omissions that could have complemented the author's analysis, such as: Tedeschi et al. "What’s the Meaning of Superhuman Performance in Today’s NLU?", ACL 2023 (a critical analysis of superhuman claims derived from using traditional benchmarks, with ideas to incentivize benchmark creators to design more solid and transparent benchmarks).

**Questions To Authors:**

Is "model metrology" really needed as a discipline? Isn't it sufficient to adhere to well known testing engineering good practices?

**Reasons To Accept:**

* contribution to the area of LLMs testing
* the idea of "model metrology" as a new discipline is appealing
* can stimulate the debate around current benchmarks limitations

**Reasons To Reject:**

* some claims not sufficiently supported
* proposal still too vague in some aspects, some ideas not so novel
* writing style

---

> ### Author Rebuttal · Authors · 2024-05-29
>
> Thank you for highlighting the need for editing. Once the paper is more concise, we will use reclaimed space to further detail the points below.
>
> While we agree HITL evals have many strong points, traditional quality assurance may be too slow for frequently retrained models. With automation we can test more thoroughly as humans generate broad scenarios rather than individual examples.
>
> Re: equating benchmarking and AGI, we offer examples throughout, eg, Bubeck et al’s “sparks of AGI” and the use of benchmarks in technical reports for GPT4, Claude, and Gemini. We will add additional support in the discussion, such as the Claude 3 announcement, which includes a y-axis labeled simply, without units, “Intelligence (Benchmark scores)”.
>
> Re: Air Canada, we agree this example doesn’t contradict the value of testing on multiple dimensions. Rather, it makes the case that we ought to expand the dimensions. While its original developers had tested the LM as a general chatbot, deployment required further stress testing by evaluators familiar with hallucinations, reasoning failures, jailbreaks, and other pitfalls.
>
> Re: novelty, many of our claims have been discussed previously, as seen in our bibliography. While we aim to give a useful overview of these issues in our paper, our main objective is to stimulate debate about the establishment of evaluation venues and a dedicated community of researchers and practitioners.
>
> Re: vagueness, While we include some proposals about how the field should work, our main thesis is that a subdiscipline is necessary. With additional space, we will expand our suggestions and add example evaluation scenarios.
>
> Question: is “model metrology” really needed?
>
> First, the incentives of our community do not reward evaluation work on par with other work (briefly addressed in sec 5.1). Second, there is no community of evaluation *practitioners* to support downstream developers. Infosec has a specialized community of pentesters who have standard practices and frameworks to creatively identify security flaws. Likewise, evaluation practitioners should have standard practices and frameworks for devising creative—but scalable—evaluations.
>
> Another reason for a specialized community is illustrated by reviewer WdB7’s invocation of the IR eval literature omitted from our paper. As long as model developers focus on their specific applications with evaluation as an afterthought, general insights into benchmarking remain siloed by application and domain.

---

> > ### Comment · Reviewer_GTUZ · 2024-06-04
> >
> > I thank the authors for their response. I still perceive some weaknesses in this work, as reported in my review. However I'm sympathetic to the idea of model metrology and slightly incrase my score, provided that the authors incorporate my comments and those of the other reviewers if accepted

---

> > ### Author Response · Authors · 2024-06-04
> > **Thank you for your response**
> >
> > Thanks for the response! We are looking forward to integrating your suggested discussions, tweaks, and references. To recap (and elaborate on the limited character count in the initial rebuttal) we plan to achieve this by:
> >
> > 1. Using the reclaimed space from conciseness edits to directly address your requests (eg., HITL evals and their shortcomings wrt our proposal) in the main text
> > 2. Clarifying that the main purpose of the metrology discipline is to bring rigor and common practices to benchmark development through a dedicated community with shared knowledge (as WdB7's complaint highlights that benchmarking knowledge is currently disorganized and rarely disseminated between subfields)
> > 3. Stating clearly that it is *very hard to make a benchmark that is both a useful tool of scientific inquiry and useful yardstick for application deployment*---but by **separating these goals** and using shared metrologist knowledge we may make strong benchmarks to achieve either goal
> > 4. In appendices, adding "full stack" hypothetical case studies describing how our desired tooling (built by metrologists) could expand specific application constraints into LM-driven (but not LM-judged) dynamic benchmarks in concrete settings
> >
> > Once again, thanks for your time and effort! Regardless of outcome, our manuscript is strengthened through your feedback.

---

### Official Review · Reviewer_WdB7 · 2024-05-12

**Rating:** 2
**Confidence:** 5
**Ethics Flag:** 1

**Summary:**

This position paper argues that there should be a division of labor between researchers developing language models, and “model metrologist” that should devote their efforts to measure the quality of LLMs wrt their actual usage.

**Reasons To Accept:**

Discussions of evaluation methodology are always a good idea (if they contain a substantial argument)

**Reasons To Reject:**

The observation that static benchmarks do not reflect real-world use cases is not new. Much of the work in information retrieval in the last 30 years has been devoted to this insight, combining research on static benchmarks with HCI research and analysis of commercial applications. There are a lot of concrete approaches in that field alone that could be discussed wrt their limitations and transferability to LLMs. The paper ignores this line of research entirely.


The paper is written as if there wasn’t plenty of discussion (including the limitations) of red teaming, LLM as a judge, and interactive evaluation of LLMs. One of the most popular benchmarks is the LMSYS Chatbot Arena, which is open-ended and interactive, but is not discussed in the submitted paper. Instruction tuning datasets are often modeled after actual usage statistics. MTBench, one of the most reported benchmarks for LLM is based on LLM as a judge, and the paper describing MTBench contains a correlation analysis to human ratings and a discussion of self-enhancement bias, but it is not even cited in the paper under review. Overall, the lack of research into evaluation literature/approaches (and/or the lack of discussion thereof) is staggering - it almost seems like the authors submitted the paper mainly to outsource the collection of relevant literature to reviewers.

---

> ### Author Rebuttal · Authors · 2024-05-29
>
> We are excited to see a spirited debate—the ultimate goal of a position paper. The polarized reviews exactly reflect the discussion we hope to prompt: Is the existing discourse on evaluation adequate for our current moment, or do we need to silo evaluation as its own specialized professional discipline? Even if you are not convinced by our argument, we hope you recognize this as a debate worth having.
>
> We discuss LM-as-judge criticisms, including several recent references (sec 4.2, 4.3.1 cite Kocmi & Federmann, 2023; Wang et al., 2023a; Mizumoto & Eguchi, 2023). However, we are happy to expand our discussion to detail the specific problems underlying these issues, such as how LMs prefer their own outputs and give biased numerical scores.
>
> Thanks for highlighting MTBench and chatbot arena as examples of modern evaluation which are important to address in the CR. Online chatbot eval actually exhibits several problems we identify: it is neither ecologically valid (tech enthusiasts on LMSys are a selective sample and user preferences change as they are exposed to more LM output) nor constrained (“chatbot quality” is open-ended). Furthermore, these HITL evaluations are not “plug-and-play” (sec 3) and require developers to build custom eval environments. Finally, relative rankings fail to provide bounded (and actionable) tolerances that can inform development. We will discuss these HITL eval drawbacks while acknowledging their advantages: there are no answers to memorize, and leaderboard rankings are unlikely to saturate.
>
> We aren’t IR experts, but the fact that you can allude to a large body of glaring omissions highlights exactly the sort of transdisciplinary “horizontal dissemination” that justifies a dedicated metrology community. Given that we already have ten pages of references, this certainly isn’t a sign of our neglect or disengagement with the evaluation literature. Practitioners certainly don’t acquaint themselves with the evaluation issues of every specialized field—they simply homebrew their own evals or rely on stale benchmarks. A metrology subdiscipline can focus on understanding eval issues across domains, freeing domain experts to focus on system-building.

---

> > ### Comment · Reviewer_WdB7 · 2024-06-04
> >
> > I read the author's response, and while I agree that spuring discussion is a worthy goal, I still think that the paper is lacking substance and overlooks too many efforts and developments already undertaken in the direction that the authors advocate for.

---

> > ### Author Response · Authors · 2024-06-04
> > **Thank you for your response**
> >
> > We will of course incorporate the helpful suggestions you have given, but it is difficult for us to incorporate or respond to other parts of your review without specific recommendations or weaknesses.
> >
> > In your view, how many citations to previous discussions and ongoing developments would be appropriate for a conference paper? The current version has 100 citations. Is there a specific crucial citation that we overlooked?
> >
> > Unfortunately, ten pages cannot encompass the current discussion on evaluation in every application, nor can it give a comprehensive view of how a specialized discipline of evaluation should engage with their work. While we have made an effort to discuss both of those topics in detail, we understand that our discussion is limited. **This is, again, the crux of our argument: modern evaluation is too specialized and too critical to leave the transmission or development of guidelines to brief papers in academic machine learning conferences.**

---

### Author Response · Authors · 2024-06-07
**Thanks and closing summary**

Thanks for your time, reviewers. Here are some parting thoughts to close out the discussion.

We appreciate your recognition of the timeliness, relevance, and importance of addressing the evaluation problem now, praising our work for “*addressing a critical current issue*” (nguQ) that is “*relevant and interesting*” (GTUZ) with “*a compelling and timely argument relevant for COLM*” (3uTB).

The reviewers broadly agreed with and recognized the novelty in our proposed solution to this problem—developing a new subdiscipline of *model metrology*. GTUZ found the idea appealing on first review, and was won over to raise their score to an accept following our rebuttal clarifying why a focused community is important. 3utb pointed out that centering “*expertise outside the field of ML*” as essential, and organizing an evaluation-centered cross-disciplinary community will enable this. nguQ pointed to the metrology idea as a core reason to accept for its “*innovative approach*” with “*the potential to revolutionize LLM assessment.*”

We are pleased to see such a broad range of perspectives shared by all reviewers. A great position paper cannot please everyone, as their purpose is to put forth debates that advance the field. Considering this, we are glad to see spirited debate in our reviewers’ positions. GTUZ directly recognized this strength, saying that the “*interesting ideas [*in our work*] can be the basis for stimulating a debate*” toward better practices for benchmarking.

This disparity in reviewer perspectives underlines why organizing a unified field is instrumental. For example, WdB7 noted that we did not put much focus on how the information retrieval community in particular has handled evaluation. Our authors’ are experts in other domains. The fact they overlooked IR illustrates why a dedicated metrology discipline is so important—if we only worked in a text generation silo, we may never have benefited from the “*30 years of devoted work*” in IR. WdB7’s excitement to share this with us will make a great case study to complement the others we planned to add following nguQ’s request in our appendix.

We appreciate all reviewer suggestions that will improve our camera ready version. While a position paper is *not* meant to provide a comprehensive literature review, and appreciate GUTZ’s compliment on the extensiveness of our 100 reference bibliography, we are excited to add discussion of the reviewer-suggested related works. Finally, we are excited to make the clarifications we arrived at in our initial responses to all reviewers.

Thank you all for your detailed and constructive feedback.

---

### Decision · Program_Chairs · 2024-07-10

**Decision:**

Accept

**Comment:**

The topic is a relevant and interesting one, and this position paper contains interesting ideas that can be the basis for stimulating a debate in that direction.
It also identifies several issues with the current approaches to evaluation/benchmarking, including benchmark saturation, lack of ecological validity, an over-focus on benchmarking generalized capabilities. The author's proposal for a new field of study - model metrology - is novel and promising.
As noted by reviewers, it would be good to:
-engage with other forms of evaluation and/or discussing elements of a user's experience/model capabilities that cannot be fully captured through the benchmarking paradigm.
-provide more concrete examples or case studies of how these benchmarks could be designed and used in practice.
-explore in more depth the practical challenges of creating and implementing dynamic benchmarks.
-add missing references.
It would be good to add these elements in the camera-ready version of the paper.

[At least one review was discounted during the decision process due to quality]